# CAF-associated genes putatively representing distinct prognosis by *in silico* landscape of stromal components of colon cancer

**Kota Okuno**[1], **Kyonosuke Ikemura**[1], **Riku Okamoto**[1], **Keiko Oki**[1], **Akiko Watanabe**[1],
**Yu Kuroda**[1], **Mikiko Kidachi**[1], **Shiori Fujino**[1], **Yusuke Nie**[1], **Tadashi Higuchi**[2],
**Motohiro Chuman**[2], **Marie Washio**[2], **Mikiko Sakuraya**[2], **Masahiro Niihara**[2],
**Koshi Kumagai**[2], **Takafumi Sangai**[3], **Yusuke Kumamoto**[4], **Takeshi Naitoh**[5], **Naoki Hiki**[2],
**Keishi Yamashita**[1]*

1 Division of Advanced Surgical Oncology, Research and Development Center for New Medical Frontiers,
Kitasato University School of Medicine, Sagamihara, Japan, 2 Department of Upper Gastrointestinal
Surgery, Kitasato University School of Medicine, Sagamihara, Japan, 3 Department of Breast and Thyroid
Surgery, Kitasato University School of Medicine, Sagamihara, Japan, 4 Department of General-Pediatric-
Hepatobiliary Pancreatic Surgery, Kitasato University School of Medicine, Sagamihara, Japan, 5 Department
of Lower Gastrointestinal Surgery, Kitasato University School of Medicine, Sagamihara, Japan

* keishi23@med.kitasato-u.ac.jp

pone.0299827

GERMANY

**Data Availability Statement:** GSE35602: https://
www.ncbi.nlm.nih.gov/geo/query/acc.cgi?acc=

## Abstract

Comprehensive understanding prognostic relevance of distinct tumor microenvironment
(TME) remained elusive in colon cancer. In this study, we performed *in silico* analysis of the
stromal components of primary colon cancer, with a focus on the markers of cancer-associ-
ated fibroblasts (CAF) and tumor-associated endothelia (TAE), as well as immunological
infiltrates like tumor-associated myeloid cells (TAMC) and cytotoxic T lymphocytes (CTL).
The relevant CAF-associated genes (CAFG)(representing R index = 0.9 or beyond with
*SPARC*) were selected based on stroma specificity (cancer stroma/epithelia, cS/E = 10 or
beyond) and expression amounts, which were largely exhibited negative prognostic
impacts. CAFG were partially shared with TAE-associated genes (TAEG)(*PLAT*, *ANXA1*,
and *PTRF*) and TAMC-associated genes (TAMCG)(*NNMT*), but not with CTL-associated
genes (CTLG). Intriguingly, CAFG were prognostically subclassified in order of fibrosis (rep-
resenting *COL5A2*, *COL5A1*, and *COL12A1*) followed by exclusive TAEG and TAMCG.
Prognosis was independently stratified by *CD8A*, a CTL marker, in the context of low
expression of the strongest negative prognostic CAFG, *COL8A1*. CTLG were comprehen-
sively identified as *IFNG*, *B2M*, and *TLR4*, in the group of low S/E, representing good prog-
nosis. Our current *in silico* analysis of the micro-dissected stromal gene signatures with
prognostic relevance clarified comprehensive understanding of clinical features of the TME
and provides deep insights of the landscape.

## Introduction

Cancer is widely recognized as a genetic disease, arising from a combination of inherited
(germline mutations) and acquired (somatic mutations) genetic aberrations. These anomalies

GSE35602 GSE17538: https://www.ncbi.nlm.nih.gov/geo/query/acc.cgi?acc=GSE17538.

**Funding:** The author(s) received no specific funding for this work.

**Competing interests:** The authors have declared that no competing interests exist.

include gene amplifications [1–3], gene losses [4–6], and gene mutations [7–11], which cumulatively lead to the transformation of cells. Interestingly, despite the accumulation of these genetic alterations, a mathematical model has suggested a relatively low selective advantage (0.004) during tumor progression [12]. Consistent with these findings, single driver gene, for example *c-MYC* expression did not show prognostic relevance at all in human colorectal cancer (CRC) [13, 14], although its deletion rescued tumorigenesis of *APC* deficiency [15]. These finding suggested that single driver gene expression alone can not phenotypically affect cancer metastasis.

On the other hand, minimal functional driver gene heterogeneity of mutations was recently confirmed among the metastasis of the individual cancer patients [16, 17], and hence tumor progression is rather associated not with genetic aberrations, plus with epigenetic [18] and/or tumor microenvironment (TME) abnormality [19]. In our current study, we therefore focused on the TME affecting patient prognosis of CRC. The TME are mainly composed of cancer-associated fibroblasts (CAF), tumor-associated endothelia (TAE), cytotoxic T lymphocytes (CTL), and tumor-associated myeloid cells (TAMC) and their derivatives or subpopulations.

The public databases of stromal expression after microdissection of the 13 CRC tumors (GSE35602) [20] was herein used for *in silico* landscape of stromal components of colon cancer together with comprehensive prognostic relevance was assessed from the bulky tissues in the 232 colon cancer patients (GSE17538) [21] to clarify their comprehensive prognostic roles of the TME. Objectives of in silico analysis of CRC tumors were to obtain results from the same assessment, and public database can be accessed and obtained by other researchers.

## Material and methods

### Expression profiles of the microdissection tissues of the 13 CRC tumors (GSE35602)

The public databases of stromal (cancer str) expression after microdissection of the 13 CRC tumors (GSE35602) were used in the microarray (Human Genome, Whole 4x44K, Agilent Inc, Santa Clara, CA) harboring 45 015 genes [20], where cancer str/epi expression ratio (cS/E ratio) was calculated. Signal intensities were adjusted by single *GAPDH* probe (A_23_P13899).

### Prognostic analysis from the bulk tissues of the 232 colon cancer tumors (GSE17538)

The public database (GSE17538) was used for prognostic analysis of 232 colon tumor tissues in the microarray base (Human Genome U133 Plus 2.0 Array, Thermo Fisher Scientific Inc, Waltham, MA) [21]. Signal intensities were adjusted by single *GAPDH* probe (212581_x_at). This database included clinicopathological factors (age, sex, T factor, N factor, M factor, and prognosis representing death). Area under curve (AUC) was calculated for prediction of death according to the individual probes, and the best optimized cut-off value was determined.

### Statistical analysis

Follow up data were evaluated in terms of overall survival. The follow-up time was calculated from the date of surgery to death or end-point. Overall survival was estimated by the Kaplan-Meier method, and compared using the log-rank test. Variables suggesting potential prognostic factors on univariate analyses were subjected to a multivariate analysis using a Cox proportional-hazards model. A P-value <0.05 was considered to indicate statistical significance. All statistical analyses were conducted using the SAS software package (JMP Pro16, SAS Institute, Cary, NC).

### Ethics statement

Ethical approval has not been obtained because this is an *in silico* study using only the Public Database.

## Results

### CAF-associated genes (CAFG) and other TME-associated genes in cancer str of the CRC tumors by *in silico* analysis

The TME is composed of CAF and TAE representing tumor angiogenesis as well as immuno-logical infiltrates such as myeloid cells, T cells, and B cells or their subpopulations. We initially identified the most relevant CAF-associated genes (CAFG, 256 probes in S1 Table) after selection by stroma specific (cS/E = 10 or beyond) and their strong association (R index = 0.9 or beyond) with well-known stromal marker representing fibroblasts [22], *SPARC* expression in cancer str of the CRC tumors after microdissection in the array-based public database (GSE35602) [20]. Intriguingly, they included well-established CAF markers including *VIM*, *ACTA2* (*aSMA*), *PDGFRB*, and *FAP* [23] (black color in S1 Table), suggesting that *SPARC* expression may represent CAF activation in the TME.

The 4 CAF markers have been all demonstrated to be pan-CAF ones, where single cell RNA sequencing (scRNA) clarified that they were expressed in all 4 CAF subpopulations (vascular CAF, matrix CAF, cycling CAF, and developmental CAF) [24]. The 161 CAFG probes with the highest expression amounts (signal intensity/GAPDHx100 = 40 or beyond in NCC210, S1 Table, red letters) included 115 SYMBOL genes (including *SPARC*), among which the top 76 genes according to expression amounts were finally selected for prognostic analysis using the public database of 232 colon cancer patients (GSE17538) [21]. Among the 76 genes, 62 genes showed negative prognostic impact (statistically significant, p<0.05), and the 27 genes were commonly shared with any TME markers-associated genes (selected by cS/E ratio = 10 or beyond and high expression amounts = 40 or beyond similarly with CAFG) (Fig 1A).

The individual TME markers and their associated genes were selected from S2 Table for *PECAM1* (*CD34*) representing TAE (ex: *PTRF*, *VIM*, *COL4A2*, *ANXA1*) [25], S3 Table for *CD8A* representing CTL (ex: *IGFBP3*, *C3*, *FBN1*, *CYBRD1*), S4 Table for *CD14* representing TAMC (ex: *DCN*, *IGFBP7*, *NNMT*, *DKK3*) [22], S5 Table for *CD3G* representing tumor infiltrating T lymphocytes (CD3 TIL)(ex: *DCN*, *SPARC*, *LUM*, *COL1A1*), S8 Table for *ARG1* representing functional myeloid derived suppressor cell (fMDSC)(ex: *A2M*, *ACTG2*, *CAV1*), S9 Table for *CD33* representing immature myeloid derived suppressor cell (iMDSC)(ex: *IGFBP7*, *VIM*, *ACTA2*, *LAMA4*), S10 Table for *FoxP3* representing regulatory T cell (Treg)(ex: *ACTG2*, *CNN1*, *MEG3*), S11 Table for *MS4A1* representing tumor infiltrating B lymphocytes (B-TIL)(ex: *NOX1*) [26], and S12 Table for *S100A9* representing tumor-associated neutrophil (TAN)(ex: *DEFB1*) [25].

In the microarray-based database, for example, one of the CAFG, *COL3A1* expression was closely associated with *SPARC* (0.89<R<0.99) among the 3 different sets (10 separate kinds of sequences/set) of microarray probes and R indexes between the 3 probe sets were ranged between 0.86 and 0.95 each other (Fig 2A), suggesting that association indicating R = 0.9 may represent same molecular relevance. We therefore thought that gene numbers of the specific TME marker-associated genes (R index = 0.9 or beyond) may represent their molecular signature impacts representing the similar stromal molecular phenotypes (Fig 2B).

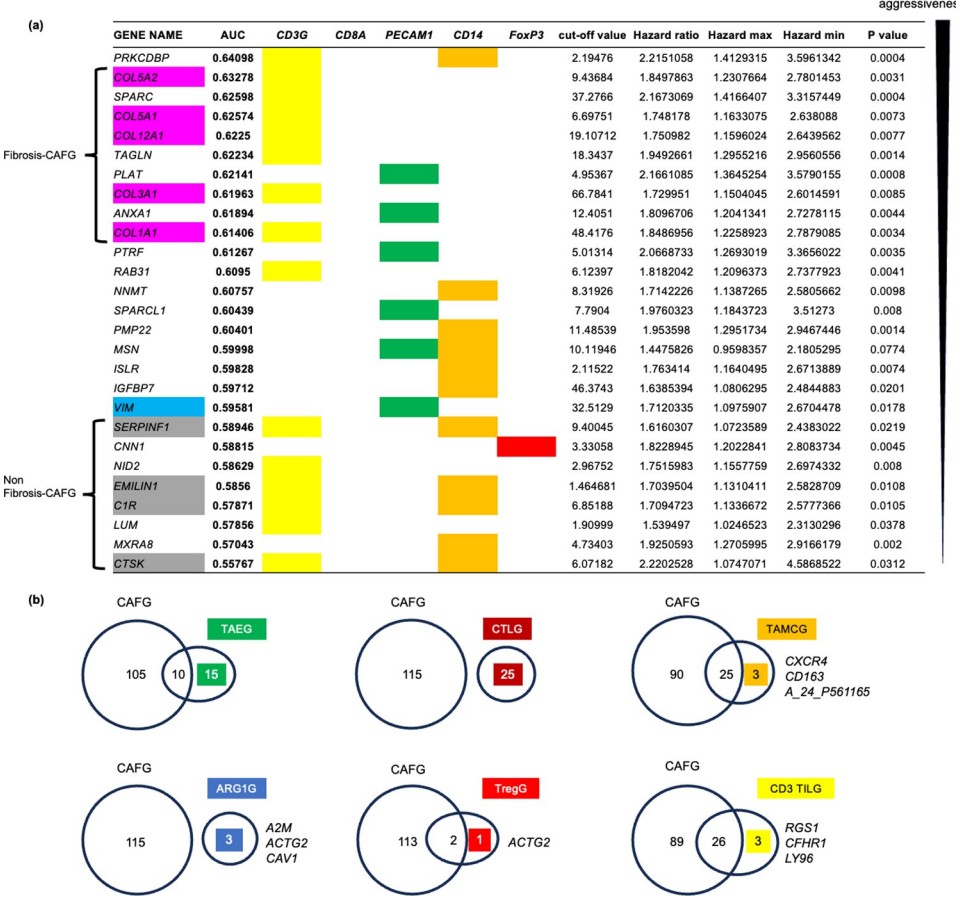

**Fig 1. Relationship between CAFG and other TME-associated genes in cancer str of the CRC tumors (GSE35602).** **(a)** The 27 CAFG that also highly (R = 0.9 or beyond) correlate with TME markers. They are listed in order of the best AUC values determined individually by prognostic analysis. In the contest of *CD3G*, fibrosis CAFG ranked as the most aggressive negative prognostic factors in contrast to non-fibrosis CAFG. **(b)** Common and unique relations to CAFG in the individual TME-associated genes.

## CAFG are partially shared with TAE-associated genes (TAEG) representing tumor angiogenesis in colon cancer

As shown in Fig 2B, *SPARC* expression putatively representing CAF activation was closely associated with 5 256 genes (R = 0.9 or beyond) (S1 Table), while *PECAM1* (*CD34*) expression represented as TAE activation was associated with 719 genes (S2 Table). Moreover, *SPARC* expression amount was much higher than that of *PECAM1* as well as other representative TME markers (in NCC210) (Fig 2C). Thus, CAF biology should play a central role in TME activation among the stromal components of the CRC tumors.

As *PECAM1* showed high cS/E ratio of 15.2 like CAFG, tumor angiogenesis was highly specific to cancer str of the CRC tumors (Fig 2D). We therefore explored genes closely associated with *PECAM1* with R = 0.9 or beyond and with cS/E = 10 or beyond (same condition with CAFG) as TAEG according to high expression amounts with signal intensity (40 or beyond, purple color in S2 Table), which identified 36 gene probes (25 SYMBOL).

Intriguingly, the 25 genes identified as TAEG were overlapped with 10 CAFG (Fig 1B), among which 7 genes were negative prognostic factors (p<0.05) and *PLAT*, *ANXA1*, and *PTRF* (*CAVIN1*) were the strongest regarding AUC (Fig 1A). On the other hand, *PECAM1*

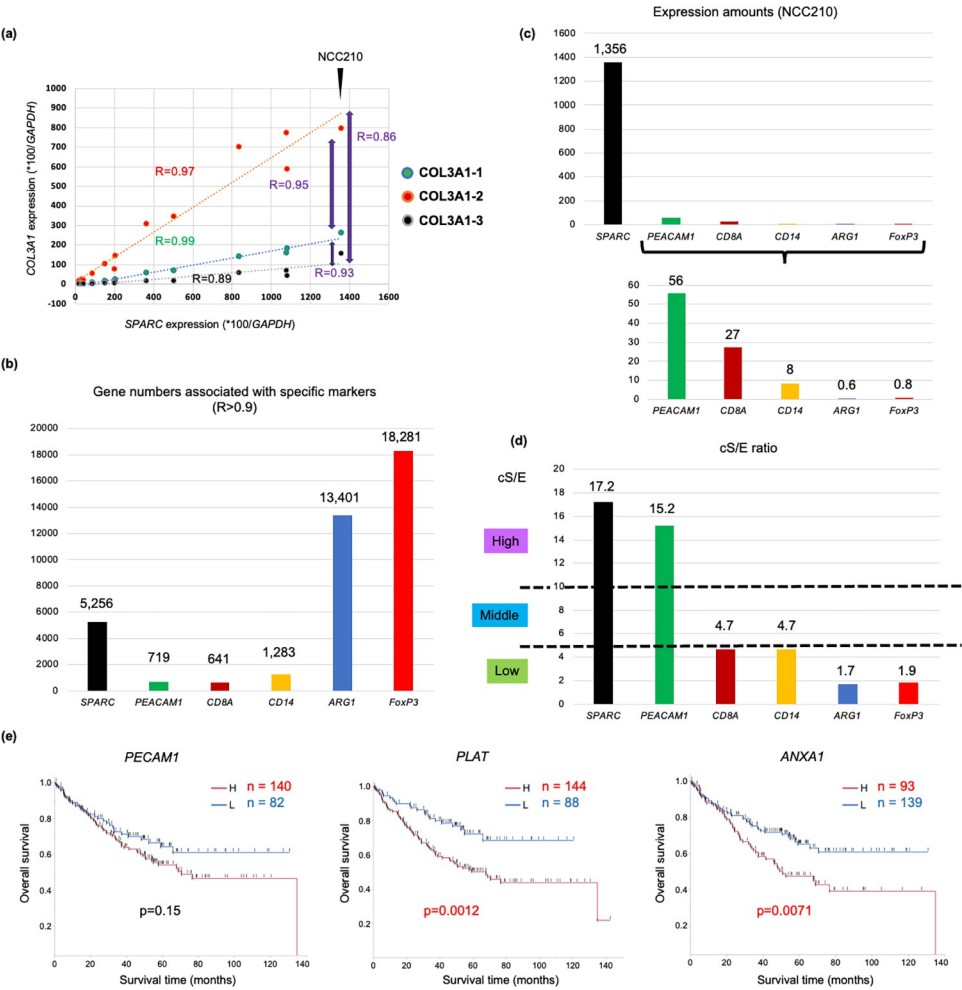

**Fig 2. TME markers and their associated group genes in cancer str of the CRC tumors (GSE35602). (a)** The 3 separate probe sets of the same gene (*COL3A1*) showed reproducible R-index near 0.90 with *SPARC*, and even between the 2 different probe sets among the 3 probes in cancer str of the CRC tumors. **(b)** The numbers of genes that closely (R>0.9) correlate with representative TME markers in cancer str of the CRC tumors. **(c)** Expression amounts of each gene in NCC210 case in cancer str of the CRC tumors. **(d)** cS/E of the representative TME markers. cS/E was classified into the 3 groups as High (cS/E = 10 or beyond), Middle (cS/E = 5 or beyond and below 10), and Low group (cS/E below 5). **(e)** High *PECAM1* expression showed poorer prognosis than its low expression in colon cancer using the best cut-off value, however there was no statistical difference (p = 0.15). On the other hand, CAFG overlapped with TAEG (*PLAT* and *ANXA1*) showed significant difference regarding prognosis in colon cancer (p = 0.0012 and p = 0.0071, respectively).

itself was not statistically significant for prognosis (p = 0.15) (Fig 2E). These findings suggested that tumor angiogenesis may be greatly affected by CAF biology, putatively because vascular CAF is the most dominant CAF subpopulation [24].

Blood vessels are composed of endothelia, pericytes, and surrounding basement membrane (BM) including type IV collagen [27] (Fig 3A), where they are marked by *PECAM1/CDH5* (*VE-cadherin*)/*TIE1*, *RGS5*, and *COL4A1/COL4A2* in the scRNA analysis [24], respectively. *PECAM1* expression was closely associated with other well-established TAE markers such as *CDH5* (R = 0.94) and *TIE1* (R = 0.93) in cancer str of the CRC tumors (Fig 3B). Pericyte-specific marker (*RGS5*) expression is also associated with *PECAM1* (R = 0.95) (Fig 3C), and BM markers (*COL4A1/COL4A2*) were tightly correlated in expression with *PECAM1* (R = 0.97),

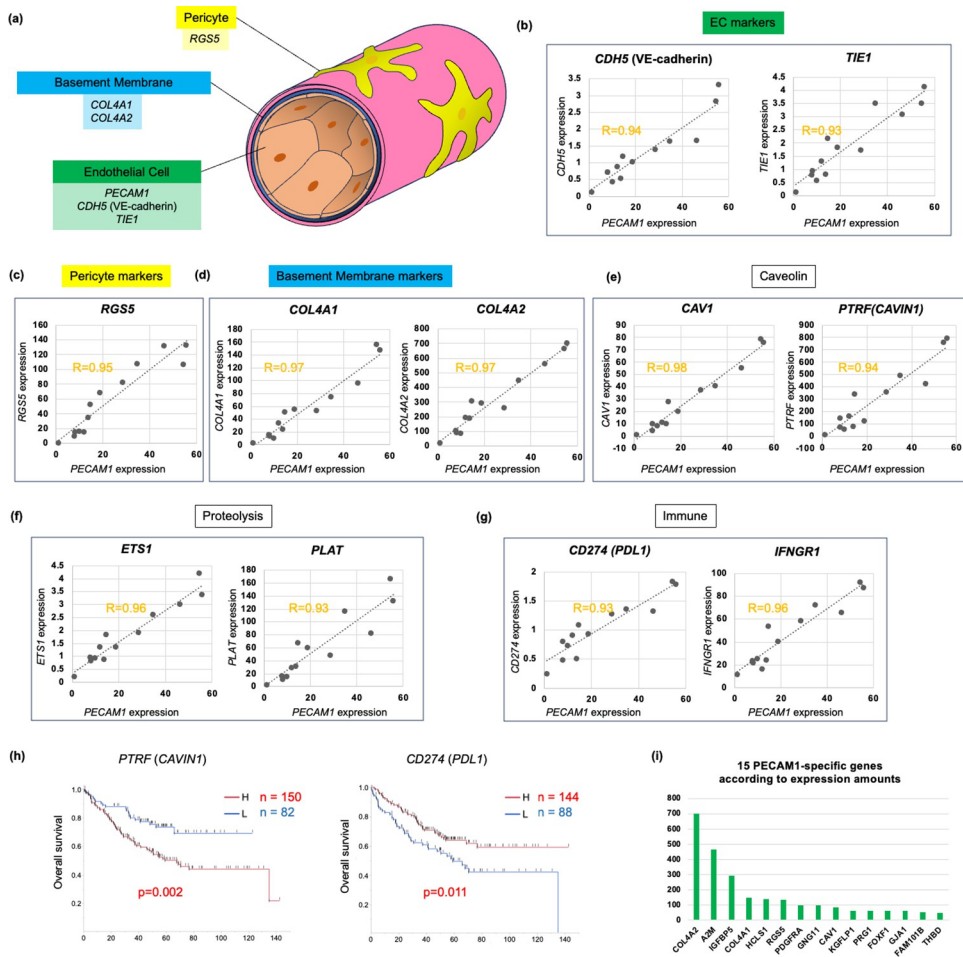

**Fig 3. TAE-associated genes (TAEG) in cancer str of the CRC tumors (GSE35602). (a)** Blood vessels are composed of endothelial cells (EC), pericytes, and the surrounding extracellular matrix of basement membrane. **(b-d)** In cancer str of the CRC tumors, *PECAM1* expression was significantly correlated with the expression of other TAE markers, such as *CDH5* and *TIE1*. **(c)** *PECAM1* expression was also closely associated with the Pericyte marker *RGS5* in cancer str of the CRC tumors. **(d)** *PECAM1* expression was closely associated with both *COL4A1* and *COL4A2*, the basement membrane components. **(e)** *CAV1* was a gene the most closely correlated with *PECAM1* expression, as was *PTRF* (*CAVIN1*), a component of caveolin in cancer str of the CRC tumors. **(f)** *PECAM1* expression was closely correlated with *ETS1/PLAT* in cancer str of the CRC tumors, which may be involved in unique proteolysis. **(g)** *PECAM1* expression was closely correlated with *CD274* (*PDL1*) / *IFNGR1* in cancer str of the CRC tumors, which may be involved in tumor immunity. **(h)** Kaplan-Meier curves for TAEG such as *PTRF* (*CAVIN1*) and *CD274* (*PDL1*) in colon cancer. **(i)** List of 15 genes of TAEG which were not overlapped with CAFG according to expression amounts of NCC210.

either (Fig 3D). These findings suggested that *PECAM1* may represent mature vascular structure in cancer str.

Among the 719 TAEG (Fig 2B), on the other hand, *PECAM1* was the most strongly associated with *CAV1* (0.97<R<0.99) in cancer str of the CRC tumors (S6 Table according to R index), suggesting that *CAV1* (12<cS/E<17) plays a critical role in tumor angiogenesis in the TME as previously shown [28, 29]. Multiple different probes for *CAV1* were enriched (10/20) as the top priority genes, which are accompanied by its close association with *PTRF* (*CAVIN1*), a component of caveola, ascribed to CAFG [30] (Fig 3E). These data indicated significant caveola contribution to tumor angiogenesis in CRC.

*PECAM1* expression was also closely associated with *ETS1/PLAT* (Fig 3F) and *CD274* (*PDL1*)/ *IFNGR1* (Fig 3G), indicating close involvement of tumor angiogenesis with unique proteolysis [31] and tumor immunity [32]. Nevertheless, their prognostic relevance was conditional; *PTRF* showed a potent negative prognostic factor putatively reflecting CAFG molecular features, whereas *PDL1* was rather a positive prognostic factor in the same database (GSE17538) (Fig 3H). Finally, the 15 unique TAEG in Fig 1B (green color) are shown according to expression amounts (Fig 3I). Intriguingly, *PDGFRA*, a matrix-CAF marker [24] was included among the TAEG.

## CAFG was prognostically independent of CTL-associated genes (CTLG) in colon cancer

The tumor infiltrates marked by *CD8A* and *CD14* were then explored as markers representing CTL and TAMC, because they were demonstrated to be their specific markers by scRNA analysis in CRC, respectively [22]. *CD8A* and *CD14* expressions were closely associated with 641 and 1 283 genes (R = 0.9 or beyond), respectively (Fig 2B), while *CD8A* expression amount was higher than *CD14* expression in this microarray database (NCC210) (Fig 2C). They were both specific to cancer str (showing cS/E = 4.7), but the values were much lower than those of *SPARC* and *PECAM1* (Fig 2D).

As expectedly, high *CD8A* expression showed significantly better prognosis than low *CD8A* expression in colon cancer (Fig 4A), as CD8 CTL was demonstrated to suppress tumorigenesis immunologically [33]. On the other hand, *CD8A* expression was not associated with a CAFG, *SPARC* except NCC210 in cancer str of the CRC tumors (R = 0.66, Fig 4B). This finding suggested that CAFG and *CD8A* may exhibit their independent contribution to prognosis, because most CAFG (75 among the 76 top CAF markers) were negative prognostic factors differently from *CD8A*. As expectedly, *CD8A* in combination with the strongest negative prognostic CAFG, *COL8A1* expression exhibited additional stratification of prognosis, especially in cases with low *COL8A1* expression (Fig 4C, both side black arrows). Interestingly, *CD8A* did not show such stratification in case of high *COL8A1* expression, suggesting that CTL effect can not overcome that of CAF for prognosis.

CTLG were initially selected similarly as 37 gene probes (25 SYMBOL genes) with cS/E ratio = 10 or beyond and expression amounts = 40 or beyond (purple colors in S3 Table), where the 25 CTLG were not overlapped with any CAFG (red brown box in Fig 1B). These findings suggested that CTL activity is independent of CAF activation (Fig 4D). On the other hand, intriguingly, *ITGB1* (cS/E ratio = 4.0) association with *CD8A* was shared with CAFG (Fig 4D), and *ITGB1* was of negative prognostic relevance (p = 0.037), suggesting that *ITGB1* may link CAF activation to CTL induction. Thus, CTL mobilization may be mediated by CAFG-associated *ITGB1* induction of T lymphocytes. On the other hand, *COL4A1/ETS1* association with *CD8A* was shared with TAEG as shown in Fig 4D, indicating that CAF activation may mediate angiogenesis with tissue destruction to CTL mobilization.

Among the top37 gene probes according to expression amounts (S3 Table), multiple probe sets of *IGFBP3* were enriched as top genes (11/37) of the CTLG (S3 Table, and Fig 5A). Although *IGFBP3* expression was strongly associated with *CD8A* expression (0.93<R<0.99) in cancer str of the CRC tumors (representative in Fig 5B), *IGFBP3* knockdown (KD) unexpectedly increased CTL mobilization accompanied by attenuated tumorigenesis [34]. These findings suggested that *IGFBP3* expression is not the cause of CTL mobilization during tumorigenesis, and may rather reflect host responses to CTL accumulation.

Consistent with the report [34], in primary colon cancer, high *IGFBP3* expression showed significantly poorer prognosis than low *IGFBP3* expression (p = 0.0028) totally differently from *CD8A* (Fig 4E). This may be partially explained by different cS/E between the 2 genes

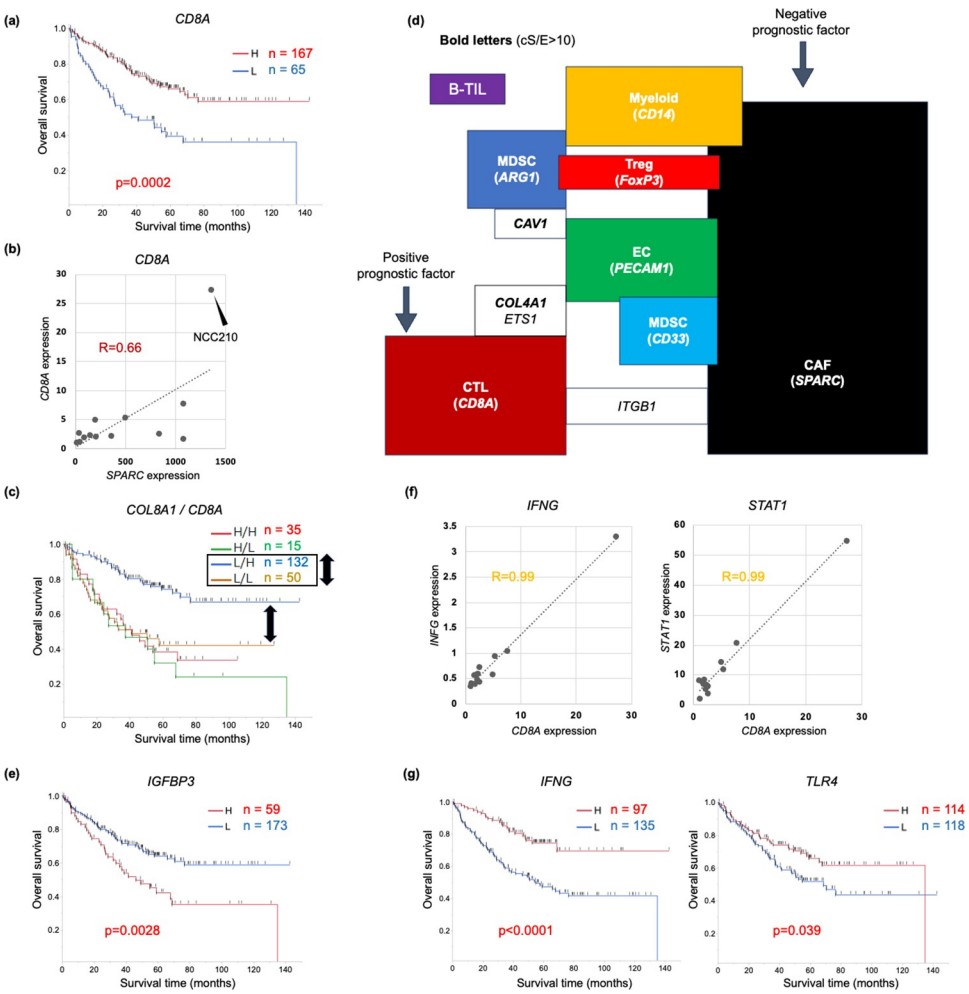

**Fig 4. Prognostic relevance of CTL marker, *CD8A*, in combination with CAFG. (a)** High *CD8A* expression showed significantly better prognosis than low *CD8A* expression in colon cancer. **(b)** *CD8A* expression did not show close association with *SPARC*, a representative CAFG except NCC210 in cancer str of the CRC tumors. **(c)** *CD8A* stratified prognosis in cases with low *COL8A1* (black double arrows), the strongest negative prognostic factor among the CAFG, while it did not stratify in those with high *COL8A1* in colon cancer. **(d)** Shema representing correlation of the individual TME-associated genes in cancer str of the CRC tumors. **(e)** Among the CTLG (cS/E = 10 or beyond, and expression amounts = 40 or beyond), *IGFBP3*, showing the highest amounts (see Fig 5A), was a strong negative prognostic factor, unlike *CD8A* in colon cancer. **(f)** *CD8A* expression was closely associated with *IFNG*, and *STAT1*, in cancer str of the CRC tumors. **(g)** High expressions of *IFNG* and *TLR4* showed significantly better prognosis than their low expression in colon cancer.

(the former was high cS/E, while the latter was low cS/E). Similarly negative prognostic effects were also confirmed for the CTLG (with high cS/E) such as *FBN1* (p = 0.0014), *CYBRD1* (p = 0.0004), *FNDC1* (p = 0.013), *MATN3* (p = 0.01), *CFH* (p = 0.022), *NOX4* (p = 0.022), *OLFML2B* (p = 0.011), and *F2R* (p = 0.012)(green bars indicating significant negative prognostic factors in Fig 5A). These findings suggested that CTLG with high cS/E have promoting role during tumor progression, and did not reflect the antitumor function of CTL.

## CTLG affecting good prognosis in colon cancer

*CD8A* expression was the most strongly associated with *IFNG* expression (0.97<R<0.99, 2.1<cS/E<2.5, S7 Table according to R index and Fig 4F), suggesting that *IFNG* could be an

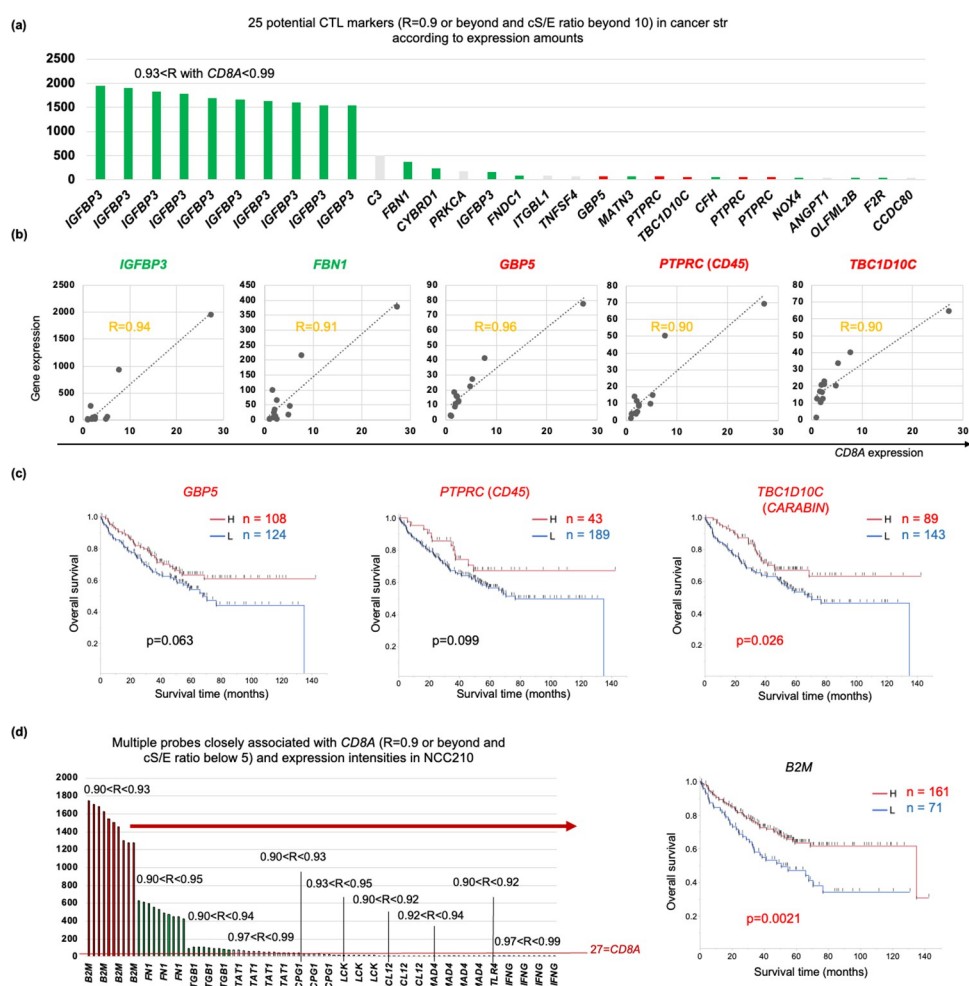

**Fig 5. CTL-associated genes (CTLG) in cancer str of the CRC tumors (GSE35602). (a)** CTLG was closely associated with a CTL marker, *CD8A* (R = 0.9 or beyond), in cancer str of the CRC tumors, and the top 25 gene probes of CTLG with cS/E = 10 or beyond are shown according to expression amounts of NCC210. **(b)** Representative genes showed correlation to *CD8A* in cancer str of the CRC tumors. Green letters represent negative prognostic factors, while red letters represent positive prognostic factors in colon cancer (GSE17358). **(c)** Survival curves of the positive prognostic CTLG with cS/E = 10 or beyond in colon cancer (GSE17358). **(d)** Representative genes of CTLG with cS/E below 5 according to expression amounts of NCC210 (left panel), and red bars (*B2M*, *TLR4*, and *IFNG*) represent positive prognostic factors in colon cancer (GSE17358). High *B2M* expression showed significantly better prognosis than its low expression in colon cancer (right panel).

excellent indicator of CTL activity as previously shown [35]. As in S7 Table, multiple 10 different probes for *IFNG* were enriched as the top association with *CD8A* as well as 11 probes for *STAT1* (R>0.97), a representative IFN-stimulated gene (ISG). S3 Table actually included other IFN pathway genes (*TLR3,4,7/IFI44L/IFIT3/BST2/CXCL10*) [36–38], suggesting that CTL activity may represent IFN pathway activation. From a prognostic point of view, however *IFNG* expression was the most likely to represent CTL activity (p<0.0001) followed by *TLR4* (p = 0.039) than *STAT1* (p = 0.09) (Fig 4G).

We further explored CTLG associated with good prognosis like *CD8A* and *IFNG*, because such genes may alternatively represent CTL activity in vivo. Among the CTLG with cS/E = 10 or beyond and high expression (40 or beyond) (Fig 5A), *GBP5*, *PTPRC* (*CD45*), and *TBC1D10C* (*CARABIN*) were identified (red bars), among which prognostic difference of only

*TBC1D10C* was statistically significant (p = 0.026) (Fig 5C). *TBC1D10C* knockout mice showed accumulated CTL like *IGFBP3* [39]. We then explored group of genes with low cS/E for CTL activation indicators (red letters in S3 Table), because *CD8A* is ascribed to the group (Fig 2D).

Among the low cS/E genes, *CD8A* expression was closely (R = 0.9 or beyond) associated with *B2M*, followed by *FN1*, *ITGB1*, *STAT1*, *CCPG1*, *LCK*, *CXCL12*, *SMAD4*, and *IFNG* as the multiple different probes (S3 Table), among which expression intensity of *B2M* was uniquely the highest (Fig 5D, left panel). Intriguingly, high *B2M* expression showed significantly better prognosis than low *B2M* expression in colon cancer (p = 0.0021, Fig 5D, right panel) similarly with *CD8A*, *IFNG* and *TLR4* (Fig 4A and 4G).

The CD8 CTL is a subpopulation of tumor infiltrating T lymphocytes (TIL), which were commonly marked by CD3 (*CD3G* was used in our study, because it was the most abundant in the microarray). CD3 TIL-associated genes (CD3 TILG) were overlapped with 26 CAFG that included COL family (S5 Table), the most potent negative prognostic factors among the CAFG (designated as fibrosis-CAFG in Fig 1A). CD3 TILG were alternately overlapped with CAFG related to TAMCG definitely from fibrosis-CAFG (non-fibrosis-CAFG in Fig 1A). These findings suggested that CD3 TIL may have heterogenous subpopulations that are prognostically distinct with or without COL family association. Thus, we did not perform further subpopulation analysis of *CD3G*.

## TAMC-associated genes (TAMCG) were partially overlapped with CAFG

In this study, we used *CD14* as a myeloid cell marker according to the previous report [24]. High *CD14* expression showed poorer prognosis than low *CD14* expression in colon cancer as expectedly, because TAMC was demonstrated to be conditionally involved in cancer promotion [40]. However, the prognostic difference was not statistically significance (p = 0.11), suggesting that TAMC contribution to prognosis may be weaker like TAE than CAF and/or CTL (Fig 6A).

TAMCG were identified as 32 gene probes (27 SYMBOL genes) (S4 Table), and intriguingly, the 27 TAMCG were overlapped with 25 CAFG, excluding *CXCR4* and *CD163*. *CD163* is an alternate well-established M2 macrophage marker, and its significance of prognostic stratification was confirmed differently from *CD14* (p = 0.027, Fig 6A). This finding may represent functional aspects of *CD163* [41, 42] in contrast to *CD14* [43], although their expression was closely associated with each other in the TME (R = 0.95, Fig 6B). Intriguingly, *CD163* expression was much higher than *CD14*.

Recent scRNA analysis revealed that *CXCR4* was expressed in tumor infiltrates including myeloid cells as well as T-cells and B-cells in the TME [26], suggesting that *CD14* TAMC uniquely may be accompanied by lymphocytes marked by *CXCR4*. Consistent with this hypothesis, non-fibrosis CAFG were commonly characterized by association with CD3 TIL and TAMC markers (Fig 1A). Prognostic relevance was not overlapped between CD3 TIL alone, TAMC alone, and both-associated CAFG (Fig 1A), suggesting that CAFG may represent subpopulations with differential immune infiltrates.

## *ARG1* and *CD33* putatively representing differential myeloid derived suppressor cells (MDSC) subpopulations unexpectedly exhibited good prognosis in colon cancer

Among the myeloid cells, MDSC inhibit tumor immunity by being mobilized from myeloid to peripheral tissues, and they can be marked by *CD33* or *arginase1* (*ARG1*), while all the cells marked by them did not strictly represent MDSC [40]. *CD33* expression represented immature

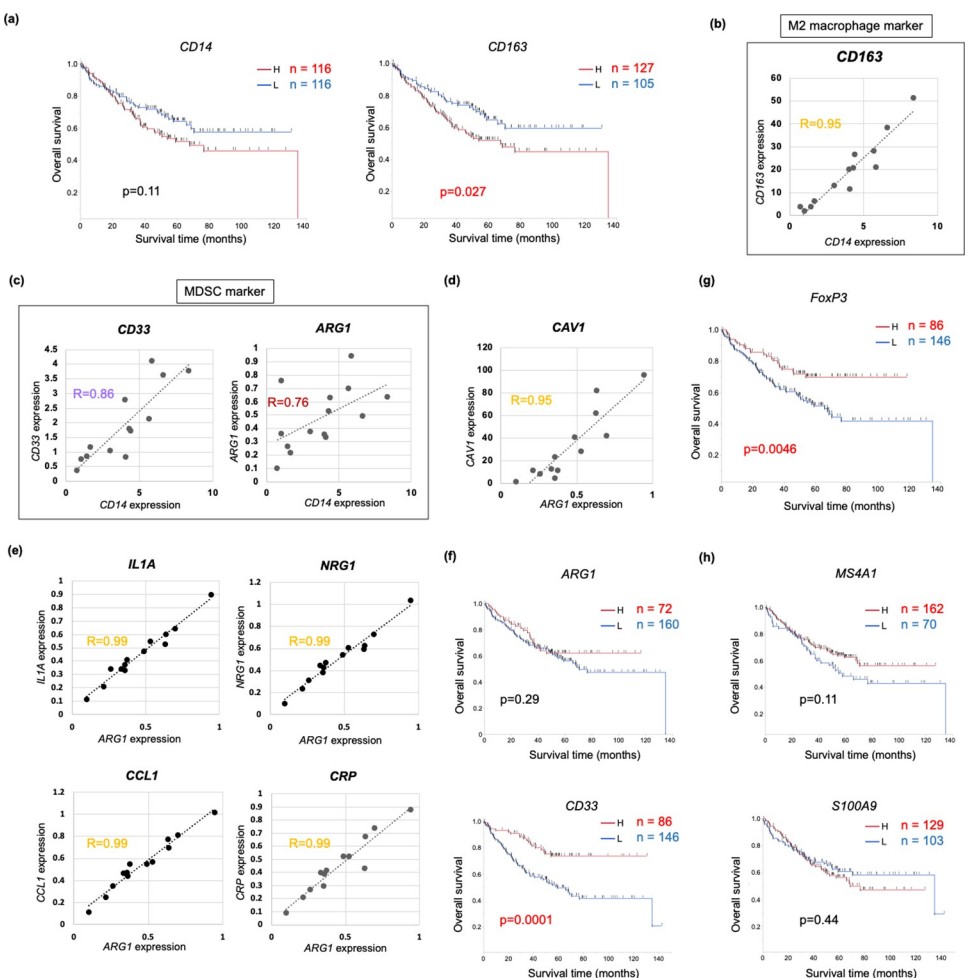

**Fig 6. Tumor-associated myeloid cells (TAMC) and other TME markers in cancer str of the CRC tumors (GSE35602). (a)** Survival curves for the TAMC marker *CD14* and the M2 macrophage marker *CD163* in colon cancer. **(b)** *CD14* and *CD163* expressions were closely associated in cancer str of the CRC tumors. **(c)** *CD33*, an iMDSC marker, and *ARG1*, a fMDSC marker, correlation with *CD14* are shown in cancer str of the CRC tumors. **(d)** *CAV1* expression was the most closely associated with *ARG1* in cancer str of the CRC tumors. **(e)** *ARG1* expression was tightly (R = 0.99) associated with expressions of inflammatory cytokines such as *IL1A* and *NRG1* including unique TAMC markers (*CCL1* and *CRP*) in cancer str of the CRC tumors. **(f)** Survival curves are shown for *ARG1* (upper panel) and *CD33* (lower panel) in colon cancer. **(g)** Survival curve is shown for *FoxP3* expression in colon cancer. **(h)** Survival curves are shown for *MS4A1*, a B TIL marker (upper panel) and *S100A9*, a TAN marker (lower panel) in colon cancer.

myeloid cells recruited from bone marrow, and showed more closely associated with *CD14* (R = 0.86), than *ARG1* (R = 0.76) did (Fig 6C). Moreover, expressions of both MDSC markers, especially *ARG1*, were much lower than *CD14* (Fig 2C). These findings suggested that *ARG1* macrophage may be small subpopulation among the TAMC.

　　*ARG1* has been demonstrated to be a fMDSC marker, because nutritional use of L-arginin, and L-arginin depletion in the TME reduces nutrition of other immunological cells such as CTL [44]. *ARG1*-associated genes (ARG1G) putatively representing fMDSC were not overlapped with any CAFG totally differently from *CD14* (Fig 1B). On the other hand, *CAV1* expression was enriched as top priority (S8 Table), and was closely associated with *ARG1* expression (Fig 6D). As a result, ARG1G were shared with TAEG representing tumor

angiogenesis (Fig 4D). This finding is consistent with the hypothesis that angiogenesis is critical for mobilization of *ARG1* macrophage from myeloid tissues into the tumor stroma.

*ARG1* expression was the most strongly associated with *IL1A*, *NRG1*, *CCL1*, *and CRP* in cancer str of the CRC tumors (Fig 6E) among the numerous associated genes (13 401 genes, Fig 2B). *CCL1* could be an aggressive TAMC marker as recently shown [45], and fluorescent double immunostaining revealed that *CCL1* with myeloid markers of *CD204* were colocalized in human cancer tissues [45]. Nevertheless, unexpectedly *ARG1* expression did not exhibit negative prognostic relevance in colon cancer (Fig 6F, top panel).

*CD33*, an alternate MDSC marker, had molecular characteristics (*CD33*-associated genes = 1 753, expression amounts in NCC210 = 3.76, and cS/E = 5.2 as shown in S9 Table), which were rather close to *CD14* than *ARG1*, and *CD33*-associated genes were shared with the most abundantly expressed CAFG, *IGFBP7*. On the other hand, *CD33*-associated genes expression was uniquely associated with angiogenic CAFG such as *VIM*, *PLAT*, and *MSN*, suggesting that CAFG-induced angiogenesis may be involved in mobilization of *CD33* macrophage. *CD33* was not significant negative prognostic factors, either, and rather a good prognostic factor (Fig 6F, lower panel), suggesting that MDSC mobilization may be insufficient to immunologically suppress tumors in the TME of clinical tumors.

## T cell subpopulations (Treg), B-TIL, and TAN and prognosis in colon cancer

*FoxP3* expression representing Treg was very low like *ARG1* expression representing fMDSC (Fig 2C) and showed low cS/E = 1.9 (Fig 2D). Both markers had a large number of the related genes (18 281 and 13 401 genes) in contrast to other TME markers (Fig 2B), and were overlapped with very few CAFG with high cS/E (S8 and S10 Tables, and Fig 1B). ARG1G were partially shared with Treg-associated genes (TregG) marked by *FoxP3*, and interestingly, many common genes between them were ascribed to low cS/E group (green box, S8 Table and S1A Fig), which included both *ATF4* and *H3F3A* as representatives. Both gene expressions were closely associated with *ARG1* expression (S1B Fig) and *FoxP3* expression (S1C Fig), respectively, in cancer str of the CRC tumors.

We identified B-TIL-associated genes as only 46 genes as close association with B-TIL marker, *MS4S1* (S11 Table), all of which did not show high or middle cS/E. In our prognostic analysis, *MS4S1* expression did not show negative prognostic relevance (Fig 6H, upper panel). TAN-associated genes were also identified as only 1 gene (*DEFB1*) as its close association with *S100A9* (S12 Table). *S100A9* expression showed negative prognostic trend, however the difference was not statistically different (Fig 6H, lower panel).

## Discussion

Although contribution of the differential TME components to patient prognosis remains elusive, our current study proposed that CAF activation represented by *COL8A1* in addition to CTL activation reflected by *CD8A* are critical determinants of prognosis in colon cancers. Endothelial marker, *PECAM1*, and myeloid marker, *CD14*, did not show such strong negative prognostic relevance, but their associated genes are partially shared with CAFG that were of prognostic importance (Fig 1A), and the shared TAEG and TAMCG may be involved in tumor aggressiveness controlled by CAFG.

scRNA analysis recently demonstrated that CAF subpopulations were composed of vascular CAF, matrix CAF, cycling CAF, and developmental CAF, and the most major component (~60%) was vascular CAF [24]. This finding suggested that CAF plays an important role in tumor angiogenesis, and the common genes between CAFG and TAEG included *PLAT*

followed by *ANXA1* and *PTRF* according to their prognostic importance (Fig 1A). Angiogenesis has been demonstrated to be promoted recently by *PLAT* [46] and classically by *ANXA1* [47]. These findings suggested that angiogenesis itself is not a potent prognostic relevance, but overlapped features of CAFG and TAEG represent aggressiveness of CRC.

The TAEG well represented tumor angiogenesis. The established TAE markers (*PECAM1*, *CDH5*, and *TIE1*) were correlated each other, accompanied by close association with pericyte marker (*RGS5*) and BM markers (*COL4A1/COL4A2*). The TAEG included genes involved in unique proteolysis (*ETS1* and *PLAT*) and immunity (*IFNGR1* and *PDL1*) as well as caveolin formation (Fig 3E–3G). Caveolin has been well known to be required for angiogenesis [28, 29], while unique proteolysis by *ETS1* was recently demonstrated to be involved in transcriptional regulation of tissue destructive genes [31] that might include *PLAT*. Interestingly, degradation of BM including type IV COL is rate-limiting step for cancer intravasation into blood in metastasis [48], and *PLAT* as well as *PLAU* may be the initial members of the protease cascade [49].

In our current study, CAFG always showed negative prognostic relevance, among which COL family genes were uniquely enriched as the most aggressive phenotypes (designated as fibrosis CAFG in this study). Previous reports suggested that CAF induce collagen fibers and fibrosis of the tissue, which hardens the ECM and contributes to malignancy, suggesting that Fibrosis-CAFG is a poor prognostic factor via the COL family [50]. Such fibrosis CAFG comprising of *COL5A2*, *COL5A1*, *COL12A1*, *COL3A1*, and *COL1A1* have recently demonstrated that they have unique molecular mechanism to CAF activation, respectively [51–53]. Moreover, fibrosis CAFG may include *SPARC* and *TAGLN* in addition to COL family genes themselves (Fig 1A), because they have been demonstrated to be involved in COL family expression induction [54, 55].

The fibrosis CAFG were shared with CD3 TILG but not with TAMCG, indicating that cancer prognosis may be greatly affected by definite components of TME. From a prognostic point of view, association with CD3 TILG alone, TAMCG alone and both may have differential CAFG molecular features, hence exhibiting different tumor immunity. In our data, *CXCR4* was included among the TAMCG, and scRNA assay recently clarified that *CXCR4* is expressed in lymphocytes as well as myeloid cells [26], suggesting that *CD14* TAMC uniquely may be accompanied by lymphocytes marked by *CXCR4*. Consistent with this hypothesis, non-fibrosis CAFG were commonly characterized by association with CD3 TIL and TAMC markers (Fig 1A).

Among the CTLG, positive prognostic factors similarly with *CD8A* were rather few, where we identified such genes as *B2M* (the highest expression amounts of 40 or beyond) followed by *TLR4* and *IFNG* (all of which belonged to low cS/E group, Fig 5D). *IFNG* is a well-established CTL activation indicator [35], and *TLR4* was demonstrated to be involved in IFN pathway activation, mediated by *IRF3* and *IRF7* [38]. Intriguingly *IFNGR1* was included among the TAEG, indicating that CTL effects may be the most greatly demanded at angiogenic sites (Fig 3G).

*B2M* truncating mutations were recently discovered in melanoma, resulting in loss of surface expression of major histocompatibility complex (MHC Class I) [56], and loss of such MHC Class I-mediated antigen presentation frequently recognized in MMR-deficient colon cancer rendered these tumors resistant to CTL-mediated tumor immunity [57]. Intriguingly, γδ T cells are proved to be effectors of immunotherapy in cancers with HLA Class I defects [58]. Our data showed that *B2M* expression was strongly associated with *CD8A* expression in the multiple probes ($0.9 < R < 0.93$, Fig 5D) not in the tumor cells, but in cancer str of the CRC tumors. The association in our study was not necessarily accompanied by MHC Class I antigen expression, suggesting that *B2M* expression may be response against CTL in stromal cells. As tumor of host sensing of *IFNG* was redundant and tumors were controlled without direct T cell cytotoxicity, multiple cell type targeted by *IFNG* should be controlled for tumor

equilibrium [59]. Thus, *IFNG* expression was more potent than *CD8A* expression itself as a prognostic factor.

In this study, we explored ARG1G marked by *ARG1* and TregG marked by *FoxP3*. They are considered to be subpopulations of TAMC and CD3 TIL, respectively and the marker genes showed very small amounts of expression as compared of *CD14* and *CD3G* expression (Fig 2C) with low cS/E ratio (Fig 2D), however such trace expressions had many related genes (Fig 2B). *ACTG2* was commonly associated with both markers (Fig 1B), although there have been no reports describing relations between both markers and *ACTG2*. On the other hand, many common genes associated with both *ARG1* and *FoxP3* were identified in gene groups with low cS/E expression ratio (S1A Fig), among which *ATF4* is of particular interest, because MDSC function was recently demonstrated to be regulated by *ATF4* [60].

MDSC was demonstrated to promote cross-tolerance in cancer by expanding Treg [61], and immune tolerance to tumors is often associated with accumulation of MDSC and an increase in the number of Treg [62]. Consistent with this, both ARG1G and TregG included common genes (S1A Fig). Moreover, Treg marked by *FoxP3* showed potent positive prognostic factor in colon cancer (p = 0.0046, Fig 6G), which recapitulated the previous report [63]. These findings suggested that Treg mobilization may be insufficient to immunologically suppress tumors in the TME of clinical tumors like MDSC, either.

Our findings clarified that that both CAFG and CTLG, but other components of the TME were dependent on either factor. Among them, CTLG may be a good marker to predict Immune checkpoint Inhibitors efficacy, while CAFG remains elusive to control, as previous reports have shown that CRC patients with CAF infiltration have a poor prognosis [64, 65]. Among CAFG, the COL family is a particularly poor prognostic factor, suggesting that it could be used as a prognostic marker.

Many common genes were identified in both CAFG and CD3 TILG, indicating a heterogeneous genetic subpopulation (Fig 1A). In previous reports, these common genes are associated with poor prognosis in colorectal cancer. For example, *COL1A1* has been reported to be linked with immune infiltrating cells [66], and *RAB31* is expressed in CAFs, contributing to the malignant potential of colorectal cancer through the secretion of *HGF* in the tumor stroma [67].

In the present analysis, the results show that CTL is more strongly related to prognosis than TAE. This is consistent with a previous report that showed a clinical prognostic benefit of immune checkpoint inhibitors over anti-VEGF antibodies [68, 69]. In addition, although no treatment to control CAF has been realized, there is a report showing that control of secretome of CAF has anti-tumor effects [70], which may contribute to the development of novel therapies in the future.

This study has limitations. The selection criteria, such as the stroma/epithelia ratio or R-index of other CAFG, may introduce bias. *SPARC* was used as a criterion for CAFG selection because *SPARC* has been identified as an important fibroblast marker in previous reports [22]. It is possible that the selection of CAFG may differ slightly if other markers are used as criteria. Future research should explore alternative criteria for a more comprehensive understanding.

In conclusion, our current *in silico* analysis of the micro-dissected stromal molecular signatures with prognostic relevance elucidated comprehensive interrelations among the TME components and provides deep insights of the beautiful molecular landscape of stromal biology.

## Supporting information

**S1 Fig. Genes commonly associated with *ARG1* and *FoxP3* in low S/E group.**
(TIF)

**S1 Table. List of CAFG.**
(XLSX)

**S2 Table. List of TAEG.**
(XLSX)

**S3 Table. List of CTLG.**
(XLSX)

**S4 Table. List of TAMCG.**
(XLSX)

**S5 Table. List of CD3 TIL-associated genes.**
(XLSX)

**S6 Table. List of genes in TAEG in order of highest correlation with *PECAM1*.**
(XLSX)

**S7 Table. List of genes in CTLG in order of highest correlation with *CD8A*.**
(XLSX)

**S8 Table. List of ARG1G.**
(XLSX)

**S9 Table. List of iMDSC-associated genes.**
(XLSX)

**S10 Table. List of TregG.**
(XLSX)

**S11 Table. List of BTIL-associated genes.**
(XLSX)

**S12 Table. List of TAN-associated genes.**
(XLSX)

## Author Contributions

**Conceptualization:** Kota Okuno, Keishi Yamashita.

**Data curation:** Kyonosuke Ikemura, Riku Okamoto, Keiko Oki, Akiko Watanabe, Yu Kuroda, Mikiko Kidachi, Shiori Fujino, Yusuke Nie.

**Formal analysis:** Tadashi Higuchi, Motohiro Chuman, Marie Washio, Mikiko Sakuraya, Masahiro Niihara, Koshi Kumagai.

**Investigation:** Kota Okuno, Tadashi Higuchi, Motohiro Chuman, Marie Washio, Mikiko Sakuraya, Masahiro Niihara, Koshi Kumagai.

**Project administration:** Keishi Yamashita.

**Supervision:** Keishi Yamashita.

**Validation:** Kota Okuno, Kyonosuke Ikemura, Riku Okamoto, Keiko Oki, Akiko Watanabe, Yu Kuroda, Mikiko Kidachi, Shiori Fujino, Yusuke Nie.

**Writing – original draft:** Kota Okuno.

**Writing – review & editing:** Takafumi Sangai, Yusuke Kumamoto, Takeshi Naitoh, Naoki Hiki, Keishi Yamashita.

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
