## [Decision Letter · Decision Letter 0]

15 Jan 2024

PONE-D-23-42477CAF-associated genes putatively representing distinct prognosis by in silico landscape of stromal components of colon cancer.PLOS ONE

Dear Dr. Yamashita,

Thank you for submitting your manuscript to PLOS ONE. After careful consideration, we feel that it has merit but does not fully meet PLOS ONE’s publication criteria as it currently stands. Therefore, we invite you to submit a revised version of the manuscript that addresses the points raised during the review process.

We look forward to receiving your revised manuscript.

Kind regards,

Chen Li, Ph.D.

Academic Editor

PLOS ONE

Journal Requirements:

Reviewers' comments:

Reviewer's Responses to Questions

**Comments to the Author**

1. Is the manuscript technically sound, and do the data support the conclusions?

Reviewer #1: Yes

Reviewer #2: Yes

2. Has the statistical analysis been performed appropriately and rigorously? 

Reviewer #1: Yes

Reviewer #2: Yes

3. Have the authors made all data underlying the findings in their manuscript fully available?

Reviewer #1: Yes

Reviewer #2: Yes

4. Is the manuscript presented in an intelligible fashion and written in standard English?

Reviewer #1: Yes

Reviewer #2: Yes

5. Review Comments to the Author

Reviewer #1: The manuscript "CAF-associated genes putatively representing distinct prognosis by in silico landscape of stromal components of colon cancer" provides an in-depth analysis of the tumor microenvironment (TME) in colon cancer. It emphasizes the prognostic relevance of cancer-associated fibroblasts (CAF) and their associated genes, exploring their interactions with other components like tumor-associated endothelia, myeloid cells, and T lymphocytes. The study utilizes in silico analysis of micro-dissected stromal gene signatures, offering novel insights into the clinical features of the TME in colon cancer. This approach could significantly enhance the understanding of colon cancer progression and prognosis, potentially guiding more effective therapeutic strategies.

However, the paper should consider and discuss some additional details. Minor revisions based on the following points are recommended for acceptance:

1. How were the cancer-associated fibroblast genes (CAFGs) selected, and could the selection criteria bias the results?

2. How do different components of the TME, such as tumor-associated endothelia and myeloid cells, interact with CAFGs, and what implications does this have for colon cancer progression?

3. The manuscript mentions the subclassification of CAFGs based on fibrosis. How does this subclassification enhance the understanding of colon cancer prognosis?

4. Given the study's findings, what are the potential clinical applications, particularly in terms of targeted therapies or prognostic markers?

Reviewer #2: This study conducted an in silico analysis of stromal components in primary colon cancer, focusing on markers of cancer-associated fibroblasts and other tumor-associated elements. Authors found that CAF-associated genes generally have a negative impact on prognosis and overlap partially with genes associated with tumor-associated endothelia and myeloid cells, but not with cytotoxic T lymphocytes genes. The research highlights the prognostic significance of various stromal elements in the tumor microenvironment of colon cancer, offering new insights into its clinical features.

Comments

1. The statement "Cancer is a genetic disease" is inappropriate, being partially correct, but it requires some clarification. Cancer is indeed caused by genetic mutations, but these mutations can be either inherited (germline mutations) or acquired (somatic mutations).

2. Objectives or aims of in silico analysis of CRC tumors were not clearly stated, what’s lacking is the rational between the experiment and the identification of CAF-associated genes.

3. The resolution of data is not high enough to allow reviewers to acquire the information and make further evaluations. For example, the texts in all Figures are not readable.

4. For the presenting of the microarray-based database, the authors are encouraged to describe the big picture or broader view prior to listing specific examples. Other data or examples may also need to be discussed briefly.

5. The authors are suggested to provide more interpretation or explanations of their results and discuss the biological impact of their findings rather than simply displaying the number of genes being identified from their experiments.

6. Many genes have been identified and mentioned in this manuscript, however more detailed information regarding the known function of the top hits is lacking. To increase the biological significance of current study, the authors can briefly hypothesize potential involvement of top genes in CRC tumor pathogenesis or therapeutic applications.

6. PLOS authors have the option to publish the peer review history of their article (what does this mean?). If published, this will include your full peer review and any attached files.

Reviewer #1: No

Reviewer #2: No

---

## [Author Response · Author response to Decision Letter 0]

5 Feb 2024

PONE-D-23-42477　　CAF-associated genes putatively representing distinct prognosis by in silico landscape of stromal components of colon cancer.

PLOS ONE

Thank you for your positive comments. I responded to your advice one by one, and revised our manuscript extensively for publication.

Reviewer #1: The manuscript "CAF-associated genes putatively representing distinct prognosis by in silico landscape of stromal components of colon cancer" provides an in-depth analysis of the tumor microenvironment (TME) in colon cancer. It emphasizes the prognostic relevance of cancer-associated fibroblasts (CAF) and their associated genes, exploring their interactions with other components like tumor-associated endothelia, myeloid cells, and T lymphocytes. The study utilizes in silico analysis of micro-dissected stromal gene signatures, offering novel insights into the clinical features of the TME in colon cancer. This approach could significantly enhance the understanding of colon cancer progression and prognosis, potentially guiding more effective therapeutic strategies.

However, the paper should consider and discuss some additional details. Minor revisions based on the following points are recommended for acceptance:

 Thank you for your positive comments.

1. How were the cancer-associated fibroblast genes (CAFGs) selected, and could the selection criteria bias the results?

[Response]

Thank you for your comment. CAFGs were selected as (1) stroma/epithelia ratio=10 or beyond in GSE35602 (microdissected CRC tumors), (2) R-index with SPARC in cancer stroma=0.9 or beyond, which were included in the original paper on page 4, line 105-109 in the revised paper. As we prioritized expression amounts, we compared the expression values of NCC210, which has the highest gene expression of the 13 cases of GSE35602. 76 genes with expression levels of 100 or beyond were further used for prognostic analysis, and 62 genes were proved to be poor prognostic factor. Such information was also included in the original paper on page 4, line 115-122. The selection of CAFG was based on SPARC among the CAFG in this study, however the results may be biased if other CAFGs were used. Therefore, such limitation was added on page 15-16, line 498-502 in the revised paper.

2. How do different components of the TME, such as tumor-associated endothelia and myeloid cells, interact with CAFGs, and what implications does this have for colon cancer progression?

[Response]

Thank you for your comment. Ben figures identified overlap between CAFG and other components of the TME as shown in Fig. 1b. Such genes may be considered to represent CAFG affecting angiogenic function like ANXA1 (CAF and TAE). In our data, their prognostic relevance was confirmed, suggesting that overlapped features of CAF and TAE represent more aggressiveness than TAE markers themselves in CRC. We have added this interpretation on page 13, line 418-420.

3. The manuscript mentions the subclassification of CAFGs based on fibrosis. How does this subclassification enhance the understanding of colon cancer prognosis?

[Response]

Thank you for your comment. Fibrosis representing COL family genes were enriched in CAFG with the highest AUC, so colon cancer with the highest COL family gene expression showed the most aggressive phenotypes, suggesting COL family gene is involved to assist cancer cells rather than protection from cancer cells. Previous reports have also supported our current data. We described such hypothesis on page 14, line 432-434. 

4. Given the study's findings, what are the potential clinical applications, particularly in terms of targeted therapies or prognostic markers?

[Response]

Thank you for your comment. Our findings clarified that both CAFG and CTLG were the potent prognostic factors, while other components of the TME were dependent on either factor. Among them, CTLG may be a good marker to predict Immune checkpoint Inhibitor efficacy, while CAFG remains elusive to control. Among CAFG, the COL family is a particularly poor prognostic factor, suggesting that they have excellent potential as prognostic biomarkers. We added this description on page 15, line 481-486.

Reviewer #2: This study conducted an in silico analysis of stromal components in primary colon cancer, focusing on markers of cancer-associated fibroblasts and other tumor-associated elements. Authors found that CAF-associated genes generally have a negative impact on prognosis and overlap partially with genes associated with tumor-associated endothelia and myeloid cells, but not with cytotoxic T lymphocytes genes. The research highlights the prognostic significance of various stromal elements in the tumor microenvironment of colon cancer, offering new insights into its clinical features.

Thank you for your pertinent comments.

Comments

1. The statement "Cancer is a genetic disease" is inappropriate, being partially correct, but it requires some clarification. Cancer is indeed caused by genetic mutations, but these mutations can be either inherited (germline mutations) or acquired (somatic mutations).

[Response]

Thank you for your pertinent comment. Cancer is a genetic disease, which was either inherited (germline mutations) or acquired (somatic mutations). We added this description on page 2, line 48-53.

2. Objectives or aims of in silico analysis of CRC tumors were not clearly stated, what’s lacking is the rational between the experiment and the identification of CAF-associated genes.

[Response]

Thank you for your comment. Objectives of in silico analysis of CRC tumors were to obtain comprehensive results from the same assessment, and public database can be accessed and obtained by other researchers. We included these objectives on page 3, line 69-70.

3. The resolution of data is not high enough to allow reviewers to acquire the information and make further evaluations. For example, the texts in all Figures are not readable.

[Response]

Thank you for your advice. We will immediately improve the quality/resolution of all the figures. Also, the resolution is poor in the integrated PDF file, but the resolution seems to be clear in the individual TIF files. We would appreciate your confirmation.

4. For the presenting of the microarray-based database, the authors are encouraged to describe the big picture or broader view prior to listing specific examples. Other data or examples may also need to be discussed briefly.

[Response]

Thank you for your suggestion. As you pointed out, some parts are difficult to comprehend, so we have added representative examples of each TME-associated genes as broader view on page 4-5, line 124-132, instead of only listing them.

5. The authors are suggested to provide more interpretation or explanations of their results and discuss the biological impact of their findings rather than simply displaying the number of genes being identified from their experiments.

[Response]

Thank you for your thoughtful feedback. We acknowledge the importance of providing a more in-depth interpretation of our results and discussing the biological implications of our findings. In response to your comments, we would like to add descriptions on page 15, lines 492-497, regarding the biological impact of our findings.

6. Many genes have been identified and mentioned in this manuscript, however more detailed information regarding the known function of the top hits is lacking. To increase the biological significance of current study, the authors can briefly hypothesize potential involvement of top genes in CRC tumor pathogenesis or therapeutic applications.

[Response]

Thank you for your comment. Some of the CAFG listed in Fig. 1a were discussed in the manuscript (PLAT and ANXA1: page13, line 415-420. COL5A2, COL5A1, COL12A1, COL3A1 and COL1A1: page14, line 434-436. SPARC and TAGLIN: page14, line 437-439). Additionally, we have provided details about certain genes (COL1A1 and RAB31) and their impacts on CRC, along with references and discussion on page 15, lines 487-491.

---

## [Decision Letter · Decision Letter 1]

16 Feb 2024

CAF-associated genes putatively representing distinct prognosis by in silico landscape of stromal components of colon cancer.

PONE-D-23-42477R1

Dear Dr. Keishi Yamashita,

We’re pleased to inform you that your manuscript has been judged scientifically suitable for publication and will be formally accepted for publication once it meets all outstanding technical requirements.

Kind regards,

Chen Li, Ph.D.

Academic Editor

PLOS ONE

Additional Editor Comments (optional):

Reviewers' comments:

Reviewer's Responses to Questions

**Comments to the Author**

1. If the authors have adequately addressed your comments raised in a previous round of review and you feel that this manuscript is now acceptable for publication, you may indicate that here to bypass the “Comments to the Author” section, enter your conflict of interest statement in the “Confidential to Editor” section, and submit your "Accept" recommendation.

Reviewer #1: All comments have been addressed

Reviewer #2: All comments have been addressed

2. Is the manuscript technically sound, and do the data support the conclusions?

Reviewer #1: Yes

Reviewer #2: Yes

3. Has the statistical analysis been performed appropriately and rigorously? 

Reviewer #1: Yes

Reviewer #2: Yes

4. Have the authors made all data underlying the findings in their manuscript fully available?

Reviewer #1: Yes

Reviewer #2: Yes

5. Is the manuscript presented in an intelligible fashion and written in standard English?

Reviewer #1: Yes

Reviewer #2: Yes

6. Review Comments to the Author

Reviewer #1: The authors well addressed my questions. I have no more questions and recommend to accept and publish.

Reviewer #2: This revision demonstrates a significant improvement; the authors have addressed all of my previous comments and concerns. I don’t have any further questions.

7. PLOS authors have the option to publish the peer review history of their article (what does this mean?). If published, this will include your full peer review and any attached files.

Reviewer #1: No

Reviewer #2: No

---

## [Editor Report · Acceptance letter]

20 Mar 2024

PONE-D-23-42477R1 

PLOS ONE

Dear Dr. Yamashita, 

I'm pleased to inform you that your manuscript has been deemed suitable for publication in PLOS ONE. Congratulations! Your manuscript is now being handed over to our production team.

Kind regards, 

on behalf of

Dr. Chen Li 

Academic Editor

PLOS ONE